# DIRECT EVOLUTIONARY OPTIMIZATION OF VARIATIONAL AUTOENCODERS WITH BINARY LATENTS

## ABSTRACT

Discrete latent variables are considered important to model the generation process of real world data, which has motivated research on Variational Autoencoders (VAEs) with discrete latents. However, standard VAE training is not possible in this case, which has motivated different strategies to manipulate discrete distributions in order to train discrete VAEs similarly to conventional ones. Here we ask if it is also possible to keep the discrete nature of the latents fully intact by applying a direct discrete optimization for the encoding model. The studied approach is consequently strongly diverting from standard VAE training by altogether sidestepping absolute standard VAE mechanisms such as sampling approximation, reparameterization trick and amortization. Discrete optimization is realized in a variational setting using truncated posteriors in conjunction with evolutionary algorithms (using a recently suggested approach). For VAEs with binary latents, we first show how such a discrete variational method (A) ties into gradient ascent for network weights and (B) uses the decoder network to select latent states for training. More conventional amortized training is, as may be expected, more efficient than direct discrete optimization, and applicable to large neural networks. However, we here find direct optimization to be efficiently scalable to hundreds of latent variables using smaller networks. More importantly, we find the effectiveness of direct optimization to be highly competitive in 'zero-shot' learning (where high effectiveness for small networks is required). In contrast to large supervised neural networks, the here investigated VAEs can, e.g., denoise a single image without previous training on clean data and/or training on large image datasets. More generally, the studied approach shows that training of VAEs is indeed possible without sampling-based approximation and reparameterization, which may be interesting for the analysis of VAE-training in general. In the regime of few data, direct optimization, furthermore, makes VAEs competitive for denoising where they have previously been outperformed by non-generative approaches.

## 1 INTRODUCTION AND RELATED WORK

Variational autoencoders (Kingma & Welling, 2014; Rezende et al., 2014) are prominent and very actively researched models for unsupervised learning. VAEs, in their many different variations, have successfully been applied to a large number of tasks including semi-supervised learning (e.g. Maaløe et al., 2016), anomaly detection (e.g. An & Cho, 2015; Kiran et al., 2018), sentence interpolation (Bowman et al., 2016), music interpolation (Roberts et al., 2018) and drug response prediction (Rampasek et al., 2017). The success of VAEs rests on a series of methods that enable the derivation of scalable training algorithms to optimize their model parameters (discussed further below). A desired feature when applying VAEs to a given problem is that their latent variables (i.e., the encoder output variables) correspond to meaningful properties of the data, ideally to those latent causes that have originally generated the data. However, many real-world datasets suggest the use of *discrete* latents as they often describe the data generation process more naturally. For instance, the presence or absence of objects in images is best described by binary latents (e.g. Jojic & Frey, 2001). Discrete latents are also a popular choice in modeling sounds; for instance, describing piano sounds may naturally involve binary latents: keys are pressed or not (e.g. Titsias & Lázaro-Gredilla, 2011; Goodfellow et al., 2013; Sheikh et al., 2014). The success of standard forms of VAEs has

consequently spurred research on novel formulations that feature discrete latents (e.g. Rolfe, 2016; Khoshaman & Amin, 2018; Roy et al., 2018; Sadeghi et al., 2019; Vahdat et al., 2019).

The objective of VAE training is the optimization of a generative data model which parameterizes a given data distribution. Typically we seek model parameters $\Theta$ of a VAE that maximize the data log-likelihood, $\mathrm{L}(\Theta) = \sum_n \log\big(p_\Theta(\vec{x}^{(n)})\big)$, where we denote by $\vec{x}^{(1:N)}$ a set of $N$ observed data points, and where $p_\Theta(\vec{x})$ denotes the modeled data distribution. Like conventional autoencoders (e.g., Bengio et al., 2007), VAEs use a deep neural network (DNN) to generate (or decode) observables $\vec{x}$ from a latent code $\vec{z}$. Unlike conventional autoencoders, however, the generation of data $\vec{x}$ is not deterministic but it takes the form of a probabilistic generative model.

For VAEs with binary latent variables, as they will be of interest here, we consider the following VAE generative model:

$$p_\Theta(\vec{z}) = \mathrm{Bern}(\vec{z}; \vec{\pi}) = \prod_h \big(\pi_h^{z_h}(1 - \pi_h)^{(1-z_h)}\big), \qquad p_\Theta(\vec{x}\,|\,\vec{z}) = \mathcal{N}\big(\vec{x}; \vec{\mu}(\vec{z}; W), \sigma^2 \mathbb{I}\big), \quad (1)$$

where $\vec{z} \in \{0, 1\}^H$ is a binary code and the non-linear function $\vec{\mu}(\vec{z}; W)$ is a DNN that outputs the mean of the Gaussian distribution. $p_\Theta(\vec{x}\,|\,\vec{z})$ is commonly referred to as *decoder*. The set of model parameters is $\Theta = \{\vec{\pi}, W, \sigma^2\}$, where $W$ incorporates DNN weights and biases. We assume homoscedasticity of the Gaussian distribution, but note that there is no obstacle to generalizing the model by inserting a DNN non-linearity that outputs a correlation matrix. Similarly, the algorithm could easily be generalized to different noise distributions should the task at hand call for it. For the purpose of this work, however, we will focus on as elementary as possible VAEs, with the form shown in Eqn. (1).

Given standard or binary-latent VAEs, essentially all learning algorithms seek to approximately maximize the log-likelihood using the following series of methods (we elaborate in the appendix):

(A) Instead of the log-likelihood, a variational lower-bound (a.k.a. ELBO) is optimized.

(B) VAE posteriors are approximated by an *encoding model*, that is a specific distribution (often Gaussian) parameterized by one or more DNNs.

(C) The variational parameters of the encoder are optimized using gradient ascent on the lower bound, where the gradient is evaluated based on sampling and reparameterization trick to obtain sufficiently low-variance and yet efficiently computable estimates.

(D) Using samples from the encoder, the parameters of the decoder are optimized using gradient ascent on the variational lower bound.

Optimization procedures for VAEs with discrete latents follow the same steps (Points A to D). However, discrete or binary latents pose substantial further obstacles in learning, mainly due to the fact that backpropagation through discrete variables is generally not possible (Rolfe, 2016; Bengio et al., 2013). In order to maintain the general VAE framework for encoder optimization, different groups have therefore suggested different possible solutions: work by Rolfe (2016), for instance, extends VAEs with discrete latents by auxiliary continuous latents such that gradients can still be computed. Work on the concrete distribution (Maddison et al., 2016) or Gumbel-softmax distribution (Jang et al., 2016) proposes newly defined continuous distributions that contain discrete distributions as limit cases. Work by Lorberbom et al. (2019) merges the Gumbel-Max reparameterization with the use of direct loss minimization for gradient estimation, enabling efficient training on structured latent spaces. Finally, work by van den Oord et al. (2017), and Roy et al. (2018) combines VAEs with a vector quantization (VQ) stage in the latent layer. Latents become discrete through quantization but gradients for learning are adapted from latent values before they are processed by the VQ stage. All methods have the goal of treating discrete distributions such that standard VAE training as developed for continuous latents can still be applied. These techniques interact during training with the standard methods (Points A-D) already in place for VAE optimization. Furthermore, they add further types of design decisions and hyper-parameters, for example parameters for annealing from softened discrete distributions to the (hard) original distributions for discrete latents.

For discrete VAEs, it may consequently be a desirable goal to investigate alternative, more direct optimization procedures that do not require a softening of discrete distributions or the use of other indirect solutions. Such a direct approach is challenging, however, because once DNNs are used to define the encoding model (Point B) standard tricks to estimate gradients (Point C) seem unavoidable. A direct optimization procedure, as is investigated here, consequently has to substantially change VAE training. For the data model (1) we will maintain the variational setting and a decoding

model with DNNs as non-linearity (Points A and D). However, we will not use an encoder model parameterized by DNNs (Point B). Instead, the variational bound will be increased w.r.t. the encoder model by using a discrete optimization approach. The procedure does not require gradients to be computed for the encoder such that discrete latents are addressed without the use of reparameterization trick and sampling approximations.

## 2 TRUNCATED VARIATIONAL OPTIMIZATION

Let us consider the variational lower bound of the likelihood. If we denote by $q_\Phi^{(n)}(\vec{z})$ the variational distributions with parameters $\Phi^{(n)}$, and by $\langle h(\vec{z}) \rangle_{q_\Phi^{(n)}} = \sum_{\vec{z}} q_\Phi^{(n)}(\vec{z}) \, h(\vec{z})$ expectation values w.r.t. to $q_\Phi^{(n)}(\vec{z})$, then the lower bound can be written as:

$$\mathcal{F}(\Phi, \Theta) = \sum_n \left\langle \log \left( p_\Theta(\vec{x}^{(n)} \mid \vec{z}) \, p_\Theta(\vec{z}) \right) \right\rangle_{q_\Phi^{(n)}} - \sum_n \left\langle \log \left( q_\Phi^{(n)}(\vec{z}) \right) \right\rangle_{q_\Phi^{(n)}}, \tag{2}$$

The general challenge for the maximization of $\mathcal{F}(\Phi, \Theta)$ is the optimization of the encoding model $q_\Phi^{(n)}$. VAEs with discrete latents add to this challenge the problem of taking gradients w.r.t. discrete latents. If we seek to avoid derivatives w.r.t. discrete variables, we have to define an alternative encoding model $q_\Phi^{(n)}$ but such an encoding has to remain sufficiently efficient. Considering prior work on generative models with discrete latents, variational distributions based on truncated posteriors offer themselves as such an alternative (Lücke & Sahani, 2008). Truncated posterior approximations have been shown to be functionally competitive (e.g. Sheikh et al., 2014; Hughes & Sudderth, 2016; Shelton et al., 2017), and they are able to efficiently train also very large scale models with hundreds or thousands of latents (e.g. Shelton et al., 2011; Sheikh & Lücke, 2016; Forster & Lücke, 2018). However, the important question for training discrete VAEs is if or how truncated variational distributions can be used in gradient-based optimization of neural network parameters. We here, for the first time, address this question noting that all previous approaches relied on closed-form (or pseudo-closed form) parameter update equations in an expectation-maximization learning paradigm.

**Optimization of the Decoding Model.** In order to optimize the parameters $W$ of the decoder DNN $\vec{\mu}(\vec{z}, W)$, the gradient of the variational bound (2) w.r.t. $W$ has to be computed. We consequently need, for any VAE, a sufficiently precise and efficient approximation of the expectation value w.r.t. the encoder $q_\Phi^{(n)}(\vec{z})$. Gradient estimation is of central importance for deep unsupervised learning, and approaches, e.g., for variance reduction of estimators have played an important role and are dedicated solely to this purpose (e.g., Williams, 1992). Reparameterization finally emerged as a key method because it allowed for sufficiently low-variance estimation of gradients based, e.g., on Gaussian middle-layer units (Kingma & Welling, 2014; Rezende et al., 2014).

For discrete VAEs, however, reparameterization requires the introduction of additional manipulations of discrete distributions that we here seek to fully avoid.

Instead of using reparameterization or variance reduction, we will compute gradients based on truncated posterior as variational distributions. A truncated posterior has the following form:

$$q_\Phi^{(n)}(\vec{z}) := \frac{p_\Theta(\vec{z} \mid \vec{x}^{(n)})}{\sum_{\vec{z}' \in \Phi^{(n)}} p_\Theta(\vec{z}' \mid \vec{x}^{(n)})} = \frac{p_\Theta(\vec{x}^{(n)} \mid \vec{z}) \, p_\Theta(\vec{z})}{\sum_{\vec{z}' \in \Phi^{(n)}} p_\Theta(\vec{x}^{(n)} \mid \vec{z}') \, p_\Theta(\vec{z}')} \quad \text{if } \vec{z} \in \Phi^{(n)}, \tag{3}$$

where for all $\vec{z} \notin \Phi^{(n)}$ the probability $q_\Phi^{(n)}(\vec{z})$ equals zero. That is, a variational distribution $q_\Phi^{(n)}(\vec{z})$ is proportional to the true posteriors in a subset $\Phi^{(n)}$, which acts as its variational parameter.

We can now compute the gradient of (2) w.r.t. the decoder weights $W$ which results in:

$$\vec{\nabla}_W \mathcal{F}(\Phi, \Theta) = -\frac{1}{2\sigma^2} \sum_n \sum_{\vec{z} \in \Phi^{(n)}} q_\Phi^{(n)}(\vec{z}) \, \vec{\nabla}_W \| \vec{x}^{(n)} - \vec{\mu}(\vec{z}, W) \|^2 . \tag{4}$$

The right-hand-side has salient similarities to the standard gradient ascent for VAE decoders. Especially the familiar gradient of the mean squared error (MSE) shows that, e.g., standard automatic differentiation tools can be applied. However, the decisive difference are the weighting factors $q_\Phi^{(n)}(\vec{z})$. Considering (3), in order to compute the weighting factors we require all $\vec{z} \in \Phi^{(n)}$ to be

passed through the decoder DNN. As all states of $\Phi^{(n)}$ anyway have to be passed through the decoder for the MSE term of (4), the overall computational complexity is not higher than an estimation of the gradient with samples instead of states in $\Phi^{(n)}$ (we elaborate in Appendix A).

To complete the decoder optimization, update equations for variance $\sigma^2$ and prior parameters $\vec{\pi}$ can be computed in closed-form (compare, e.g., Shelton et al., 2011) and are given by:

$$\sigma^{2,\text{new}} = \frac{1}{DN} \sum_n \sum_{\vec{z} \in \Phi^{(n)}} q_{\Phi}^{(n)}(\vec{z}) \, \|\vec{x}^{(n)} - \vec{\mu}(\vec{z}, W)\|^2 \,, \quad \vec{\pi}^{\text{new}} = \frac{1}{N} \sum_n \sum_{\vec{z} \in \Phi^{(n)}} q_{\Phi}^{(n)}(\vec{z}) \, \vec{z}, \quad (5)$$

where $N$ is the number of samples in the training dataset and $D$ is the number of observables.

**Optimization of the Encoding Model.** After having established that the decoder can be optimized efficiently and by using standard DNN methods, the important question is if the encoder can be trained efficiently. Encoder optimization is usually based on a reformulation of the variational bound (2) given by:

$$\mathcal{F}(\Phi, \Theta) = \sum_n \left\langle \log\left(p_{\Theta}(\vec{x}^{(n)} \mid \vec{z})\right) \right\rangle_{q_{\Phi}^{(n)}} - \sum_n D_{\text{KL}}\left(q_{\Phi}^{(n)}(\vec{z}), p_{\Theta}(\vec{z})\right). \quad (6)$$

Centrally for this work, truncated posteriors allow a specific alternative reformulation of the bound that enables efficient optimization. The reformulation recombines the entropy term of the original form (2) with the first expectation value into a single term, and is given by (see Lücke, 2019, for details):

$$\mathcal{F}(\Phi, \Theta) = \sum_n \log\left(\sum_{\vec{z} \in \Phi^{(n)}} p_{\Theta}(\vec{x}^{(n)} \mid \vec{z}) \, p_{\Theta}(\vec{z})\right). \quad (7)$$

Thanks to the simplified form of the bound, the variational parameters $\Phi^{(n)}$ of the encoding model can now be sought using direct discrete optimization procedures. More concretely, because of the specific form (7), pairwise comparisons of joint probabilities are sufficient to maximize the lower bound: if we update the set $\Phi^{(n)}$ for a given $\vec{x}^{(n)}$ by replacing a state $\vec{z}^{\text{old}} \in \Phi^{(n)}$ with a state $\vec{z}^{\text{new}} \notin \Phi^{(n)}$, then $\mathcal{F}(\Phi, \Theta)$ increases if and only if:

$$\log\left(p_{\Theta}(\vec{x}^{(n)}, \vec{z}^{\text{new}})\right) > \log\left(p_{\Theta}(\vec{x}^{(n)}, \vec{z}^{\text{old}})\right). \quad (8)$$

To obtain intuition for the pairwise comparison, consider its form when inserting the binary VAE (1) into the left- and right-hand sides. Eliding terms that do not depend on $\vec{z}$ we obtain:

$$\widetilde{\log p_{\Theta}}(\vec{x}, \vec{z}) = -\|\vec{x} - \vec{\mu}(\vec{z}, W)\|^2 - 2\sigma^2 \sum_h \tilde{\pi}_h z_h \quad (9)$$

where $\tilde{\pi}_h = \log\left((1 - \pi_h)/\pi_h\right)$. The expression assumes an even more familiar form if we restrict ourselves for a moment to sparse priors $\pi < \frac{1}{2}$, i.e., $\tilde{\pi} > 0$. Criterion (8) then becomes:

$$\|\vec{x}^{(n)} - \vec{\mu}(\vec{z}^{\text{new}}, W)\|^2 + 2\sigma^2 \tilde{\pi} \, |\vec{z}^{\text{new}}| \quad < \quad \|\vec{x}^{(n)} - \vec{\mu}(\vec{z}^{\text{old}}, W)\|^2 + 2\sigma^2 \tilde{\pi} \, |\vec{z}^{\text{old}}| \quad (10)$$

where $|\vec{z}| = \sum_{h=1}^H z_h$. Such functions are routinely encountered in sparse coding or compressive sensing (Eldar & Kutyniok, 2012): for each set $\Phi^{(n)}$ we seek those states $\vec{z}$ that are reconstructing $\vec{x}^{(n)}$ well while being sparse ($\vec{z}$ with few non-zero bits). For VAEs, $\vec{\mu}(\vec{z}^{\text{new}}, W)$ is a DNN and as such much more flexible in matching the distribution of observables $\vec{x}$ than can be expected from linear mappings. Furthermore, criteria like (10) usually emerge for maximum a-posteriori (MAP) training in sparse coding (Olshausen & Field, 1996). In contrast, we here seek a *population* of states $\vec{z}$ in $\Phi^{(n)}$ for each data point. It is a consequence of the reformulated lower bound (7) that it remains optimal to evaluate joint probabilities (as for MAP) although the constructed population of states $\Phi^{(n)}$ can capture (unlike MAP training) a rich posterior structure.

But how can new states $\vec{z}^{\text{new}}$ that optimize $\Phi^{(n)}$ be found efficiently in high-dimensional latent spaces? Random search and search by sampling has recently been explored for elementary generative models (Lücke et al., 2018). Here we will follow another recent suggestion (Guiraud et al., 2018) and make use of a search based on evolutionary algorithms (EAs). In this setting we interpret sets $\Phi^{(n)}$ as populations of binary genomes $\vec{z}$ and base the fitness function on Eqn. (9).

Concretely, using $\Phi^{(n)}$ as initial parent pool, we apply the following genetic operators in sequence: firstly, *parent selection* picks $N_p$ states from the parent pool. In our numerical experiments we used

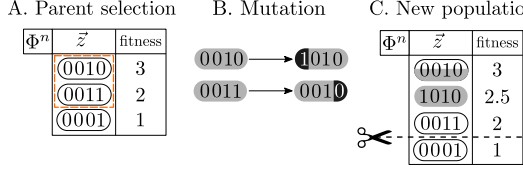

A. Parent selection    B. Mutation    C. New population

Figure 1: The optimization process of the variational parameters $\Phi^{(n)}$ using evolutionary search. **A.** Some states are selected as parents. **B.** Each child undergoes mutation. **C.** Children are merged with the original population and the least fit are discarded.

fitness-proportional parent selection, for which we add an offset (constant w.r.t. $\vec{z}$) to the fitness values in order to make them strictly non-negative. Each of the children undergoes *mutation*: one or more bits are flipped to further increase offspring diversity. In our experiments we perform random uniform selection of the bits to flip. Crossover could also be employed to increase offspring diversity. We repeat the procedure using the children generated this way as the parent pool, giving birth to multiple *generations* of candidate states. Finally, we update $\Phi^{(n)}$ by substituting individuals with low fitness with candidates with higher fitness. The whole procedure can be seen as an evolutionary algorithm with perfect memory or very strong elitism (individuals with higher fitness never drop out of the gene pool). Note that the improvement of the variational lower bound depends on generating as many as possible *different* children with high fitness over the course of training.

We point out that the EAs optimize each $\Phi^{(n)}$ independently, so this technique can be applied to large datasets in conjunction with stochastic or batch gradient descent on the model parameters $\Theta$: it does not require to keep the full dataset (or all sets $\Phi^{(n)}$) in memory at a given time. Fig 1 shows how EAs produce new states that are used to update each set $\Phi^{(n)}$. The full training procedure for binary VAEs is summarized in Algorithm 1.

---

**Algorithm 1** Training Truncated Variational Autoencoders

---

Initialize model parameters $\Theta = \{W, \vec{\pi}, \sigma^2\}$
Initialize each $\Phi^{(n)}$ with $S$ distinct latent states
**repeat**
  **for all** batches in dataset **do**
    **for** sample $n$ in batch **do**
      $\Phi^{new} = \Phi^{(n)}$
      **for all** generations **do**
        $\Phi^{new} = $ mutation (crossover (selection ($\Phi^{new}$)))
        $\Phi^{(n)} = \Phi^{(n)} \cup \Phi^{new}$
      **end for**
      Truncate $\Phi^{(n)}$ to $S$ fittest elements based on (9)
    **end for**
    Use Adam to update $W$ using objective (4)
  **end for**
  Use (5) to update $\vec{\pi}, \sigma^2$
**until** parameters $\Theta$ have sufficiently converged

---

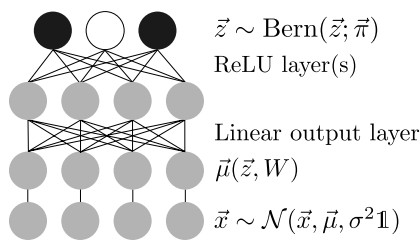

$\vec{z} \sim \text{Bern}(\vec{z}; \vec{\pi})$
ReLU layer(s)

Linear output layer
$\vec{\mu}(\vec{z}, W)$

$\vec{x} \sim \mathcal{N}(\vec{x}, \vec{\mu}, \sigma^2 \mathbb{1})$

Figure 2: Graphical representation of the model architecture used in numerical experiments.

## 3 NUMERICAL EXPERIMENTS

Having defined the training procedure, we numerically investigated its properties. After first verifying that the procedure can recover generating parameters using ground-truth data (see Appendix B), we conducted experiments to address the following two standard questions:

(1) How efficient, i.e. how scalable, is the direct discrete optimization of binary VAEs?
(2) How effective is the procedure, i.e., how well does it perform for a given VAE model?

In all numerical experiments, the training of the DNN parameters based on (4) is performed with mini-batches, the Adam optimizer (Kingma & Ba, 2014) and decaying or cyclical learning rate scheduling (Smith, 2017). Xavier/Glorot initialization (Glorot & Bengio, 2010) is used for the DNN weights, while biases are always zero-initialized. Parameters $\vec{\pi}$ and $\sigma^2$ are updated via Eqn. (5) and initialized to $\frac{1}{H}$ ($H$ is the size of $\vec{\pi}$) and 0.01 respectively. Hyper-parameter optimization was conducted manually and, for the more complex datasets, it also made use of black box Bayesian op-

Table 1: Denoising performance in PSNR (dB) for the 'house' image under controlled conditions ($D$=8×8, $H$=64 for all algorithms).

|  | $\sigma$=15 | $\sigma$=25 | $\sigma$=50 |
|---|---|---|---|
| MTMKL | **34.29** | 31.88 | 28.08 |
| GSC | 32.68 | 31.10 | 28.02 |
| VAR-BSC | 32.25 | 31.15 | 28.62 |
| TVAE | $34.27 \pm .02$ | $\mathbf{32.65 \pm .06}$ | $\mathbf{29.61 \pm .02}$ |

timization based on Gaussian Processes (Nogueira, 2019). We will refer to the binary VAE trained with the method described above as *Truncated Variational Autoencoder* (TVAE) as the use of truncated posteriors is the main distinguishing feature.

**Scalability and improvement on linear models.** Let us first numerically investigate scalability properties of TVAE especially in comparison with linear models. After verifying parameter recovery for ground-truth data (see Appendix B), we used natural data in the form of image patches as an intermediately large scale and natural dataset. Concretely, we used 100,000 whitened image patches of $16 \times 16$ pixels extracted from a standard image database (van Hateren & van der Schaaf, 1998) and pre-processed as in Guiraud et al. (2018).

The simplest possible VAEs would use linear mappings for the decoder $\vec{\mu}(\vec{z}, W)$. For standard Gaussian latents, a linear VAE can recover probabilistic PCA solutions (e.g. Lucas et al., 2019). For Bernoulli latents, we would recover binary sparse coding (Haft et al., 2004; Shelton et al., 2011) solutions. We therefore start training (using $H = 300$ latents) with a linear VAE. After 100 epochs the weights of the linear mapping were used to initialize the bottom layer of a deeper decoder network with three layers of 300, 300 and $16 \times 16 = 256$ units. The weights of the deeper layers were simply initialized to the identity matrix. Furthermore, prior and variance were optimized. The described setup guarantees a common starting point for linear and non-linear VAEs such that the difference provided by deeper decoder DNNs can be highlighted. Fig. 3 shows the variational bounds during learning of the linear VAE compared to the non-linear VAE for a typical experiment. The non-linear VAE can be observed to quickly and significantly optimize the lower bound beyond a linear VAE. We will later (when we are not interested in comparisons to linear VAEs) simply optimize the weights of non-linear TVAE directly as we did not observed an advantage by first optimizing a linear VAE.

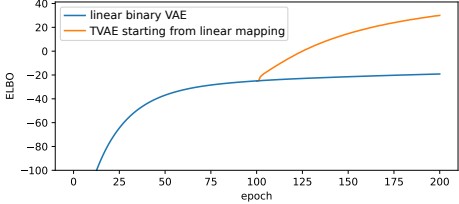

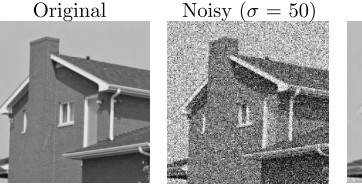

Original     Noisy ($\sigma = 50$)     Denoised

Figure 3: ELBO gain of TVAE compared to linear VAE with binary latents (on $16 \times 16$ image patches).

Figure 4: TVAE denoising of house image with noise level $\sigma = 50$. The denoised image has PSNR=30.03, the best of the runs of Tab. 2.

Compared to shallow linear models, we observed a similar efficiency and scalability of TVAE. The main additional computational costs are given by passing the latent states through the a full decoder DNN instead of just through a linear mapping. The sets of states used could be kept small, at size $S = |\Phi^{(n)}| = 64$, such that $N \times (|\Phi^{(n)}| + |\Phi^{(n)}_{\text{new}}|)$ states had to be evaluated for each epoch. This compares to $N \times M$ states that would be used for standard VAE training (given $M$ samples are drawn per data point). Differently to standard VAE training the $\Phi^{(n)}$ have to be remembered across iterations. For very large datasets, the additional $\mathcal{O}(N \times |\Phi^{(n)}| \times H)$ memory demand can be distributed over compute nodes, however. To further investigate scalability, we went to up to $H$=1000 latent variables (while using 100 units in the DNN middle layer). TVAE training time remained in line with the theoretical linear scaling with $H$ while the variational bound further increased.

**Effectiveness: Image Denoising.** As we have observed, scaling to large latent spaces does not pose a problem for the presented approach. It is clear, however, that memory and computational cost

Table 2: Denoising performance in PSNR (dB) for the 'house' image for different algorithms with different optimized hyper-parameters. The **top** category only requires the noisy image. The **middle** requires additional information such as noise level (KSVD, WNNM, BM3D) or additional noisy images with matched noise level (n2v$^\dagger$). The **bottom** requires large clean datasets.

|  | $\sigma$=15 | $\sigma$=25 | $\sigma$=50 |
|---|---|---|---|
| N2V$^\star$ | 32.05 | 29.20 | 25.42 |
| MTMKL | **34.29** | 31.88 | 28.08 |
| GSC | 33.78 | 32.01 | 28.35 |
| S5C | 33.50 | 32.08 | 28.35 |
| VAR-BSC | 33.50 | 32.32 | 28.91 |
| TVAE | $34.27 \pm .02$ | $\mathbf{32.65 \pm .06}$ | $\mathbf{29.98 \pm .05}$ |
| N2V$^\dagger$ | 33.91 | 32.10 | 28.94 |
| KSVD | 34.32 | 32.15 | 27.95 |
| WNNM | 35.13 | 33.22 | 30.33 |
| BM3D | 34.94 | 32.86 | 29.37 |
| EPLL | 34.17 | 32.17 | 29.12 |
| BDGAN | 34.57 | 33.28 | 30.61 |
| DPDNN | **35.40** | **33.54** | **31.04** |

increase with the number of data points. Above, we processed $100,000$ data points which is still feasible for the small DNNs used. However, larger DNNs increase computational load significantly because $N \times (|\Phi^{(n)}| + |\Phi^{(n)}_{\mathrm{new}}|)$ latent states have to be passed through the decoder. Furthermore, larger DNNs require more data points to not overfit which further increases computational load of our $N$-dependent method. In many applications, there is, however, anyway relatively few data available which makes the application of large DNNs prohibitive. One example is the task of 'zero-shot' denoising, i.e., denoising of an image when only the image itself is available. Learning without clean data recently became very popular. The task is currently addressed using approaches based on standard feed-forward DNNs whose training objectives have been altered to allow for training on noisy images (e.g. Lehtinen et al., 2018; Krull et al., 2019). Deep generative models are, on the other hand, more naturally suited for training on noisy data as their learning objective can be used directly. Shocher et al. (2018) also argue that smaller DNNs are sufficient for the 'zero-shot' setting. Because of its recent popularity and suitability for approaches with smaller DNNs, we consequently focus on 'zero-shot' denoising. As a very significant additional benefit, the task allows for directly comparing the VAE approach to a large range of other approaches that have recently been suggested. Most notably we can compare to non-deep generative models, large feed-forward DNNs (Zhu et al., 2019; Dong et al., 2019) and DNNs dedicated to learning from noisy data (Lehtinen et al., 2018; Krull et al., 2019).

The one denoising benchmark that offers the broadest possible comparison to other methods is probably the 'house' image (Fig. 4 left). The standard benchmark settings for 'house' make use of additive Gaussian white noise with standard deviations $\sigma \in \{15, 25, 50\}$. First, consider the comparison in Tab. 1 where all models used the same patch size of $D = 8 \times 8$ pixels and $H = 64$ latent variables (Appendix B for details). Tab. 1 lists the different approaches in terms of the standard measure of peak signal-to-noise ratio (PSNR). Values for MTMKL (Titsias & Lázaro-Gredilla, 2011), GSC (Sheikh et al., 2014) and S5C (Sheikh & Lücke, 2016) were taken from the respective original publications (which all established new state-of-the-art results when first published). As can be observed, TVAE significantly improves performance for high noise levels. TVAE is able to learn the best data representation for denoising and represents the state-of-the-art in this controlled setting (i.e., fixed $D$ and $H$). The decoder DNN of TVAE provides the decisive performance advantage: TVAE significantly improves performance compared to the linear Binary Sparse Coding (var-BSC, Henniges et al., 2010; Shelton et al., 2011), confirming that the high lower bounds of TVAE on natural images translate into improved performance on a concrete benchmark. For $\sigma = 25$ and $\sigma = 50$, TVAE also significantly improves on MTMKL, GSC, and S5C. These three approaches are based on a spike-and-slab sparse coding model (also compare Goodfellow et al., 2012). Despite the less flexible Bernoulli prior, the decoder DNN of TVAE provides the highest PSNR values for high noise levels.

In order to further extend our comparison, in the last experiment we considered the denoising task without controlling for equal conditions. Concretely, we allowed for any approach that performs denoising on the benchmark including approaches that are trained on large image datasets and/or use different patch sizes (including multi-scale and whole image processing). Note that different approaches may employ very different sets of hyper-parameters that can be optimized for denoising

performance: for sparse coding approaches, hyper-parameters include patch and dictionary sizes; for DNN approaches they include all network and training scheme hyper-parameters. By allowing for comparison in this less controlled setting, we can include a number of recent approaches including large DNNs trained on clean data and training schemes specifically targeted to noisy training data. Tab. 2 shows the denoising performance for the three noise levels we investigated, with results for other algorithms taken from their corresponding original publications unless specified otherwise. For WNNM and EPLL we cite values from Zhang et al. (2017). The results reported for noise2void (n2v, Krull et al., 2019) were produced specifically for this work (see Appendix B).

Note that the best performing approaches in Tab. 2 cannot be trained on noisy data: EPLL (Zoran & Weiss, 2011), BDGAN (Zhu et al., 2019) and DPDNN (Dong et al., 2019) all make use of clean training data (typically hundreds of thousands of data points or more). For denoising, EPLL also requires the ground-truth noise level of the test image. Ground-truth noise level information is also required by KSVD (Elad & Aharon, 2006) and WNNM (Gu et al., 2014). As noisy data is very frequently occurring, removing the requirement of clean data has been of considerable interest with, e.g., approaches like noise2noise (n2n, Lehtinen et al., 2018) and noise2void being very actively discussed currently. The n2n approach can achieve denoising performance on noisy training data which is almost as high as the performance of a given DNN when trained on clean data. It would thus outperform all approaches in Tab. 2 except for the bottom three. However, n2n requires different noise realizations of the very same underlying image. noise2void aims to remove this artificial assumption. Considering Tab. 2, PSNR values of TVAE were consistently higher than those of n2v even if n2v was trained on external data with matched-noise level (n2v$^{\dagger}$ in Tab. 2). Performance of TVAE is 0.2dB lower than BM3D for $\sigma = 25$ and 0.6dB higher for $\sigma = 50$, which makes it, for large noise levels, the state-of-the-art on this benchmark in the 'zero-shot' setting (i.e., the setting n2n and n2v aim to address).

## 4 DISCUSSION

We investigated a novel way to train VAEs with binary latents. In order to avoid derivatives w.r.t. stochastic discrete latents, we here changed the standard training setup substantially. Updates of the decoder DNN now involve a weighted sum over states (4) and the encoder DNN is replaced by a discrete evolutionary optimization. The direct optimization of the encoder replaces methods that are usually considered indispensable for the training of VAEs: sampling approximation and reparameterization trick. Furthermore, the here investigated encoding model does not use a joint mapping for all datapoints to latent space, i.e., the approach is not amortized. While amortization can be advantageous as information can be shared across datapoints, disadvantages in terms of less tight lower bounds have also been pointed out (e.g. Kim et al., 2018; Cremer et al., 2018). Related to this point, standard VAE training usually involves factored Gaussian approximations of VAE posteriors which can introduce biases (e.g., discussion by Vértes & Sahani, 2018). The investigation of alternatives may therefore, more generally, shed further light on the consequences of specific approximation choices used to define VAE encoders.

The price we pay for not using amortization is efficiency: we optimize variational parameters for each data point which is, of course, more costly. However, direct optimization can scale VAEs to large latent spaces if smaller DNNs are used. When the use of large DNNs is anyway prohibitive because of limited data, the here studied approach can play out its effectiveness. For the recently popular task of 'zero-shot' denoising, we observed state-of-the-art results in a domain where VAEs have not been reported to be competitive before. The competitive performance is presumably due to the approach not being subject to an amortization gap as well as not being based on factored variational distributions.

Our conclusion is consequently that direct discrete optimization can serve as an alternative for training discrete VAEs. In a sense, the approach can be considered as a brute-force optimization which is slower than conventional amortized training but more effective for scales at which it can be applied. To our knowledge, the approach is also the first training method for VAEs that is not using sampling-based gradient estimates, and the first which makes VAEs competitive for 'zero-shot' denoising.

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

# A    DETAILS OF ENCODER AND DECODER OPTIMIZATION

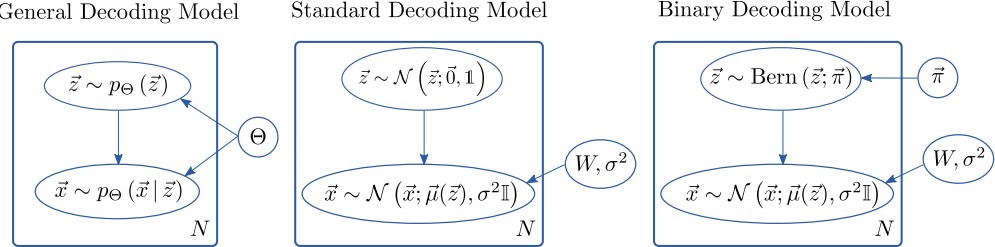

Figure 5: From left to right: generic VAE decoding model, continuous-latent VAE model with Gaussian noise and the binary-latent VAE model of Eqn. (1), in plate notation.

See Fig. 5 for a graphical comparison between the decoding models of a vanilla VAE and the binary VAE considered here (1). Fig. 6 graphically illustrates different steps to optimize standard VAEs, and additional steps suggested by different contributions in order to optimize discrete VAEs.

For the optimization of the binary VAE (1), consider the original form of the lower bound, Eqn. (2). When taking derivatives of $\mathcal{F}(\Phi, \Theta)$ w.r.t. $\Theta$ we can ignore the entropy term[1]. For the binary VAE model of Eqn. (1) the gradient of the lower bound w.r.t. $W$ is then given by:

$$
\begin{aligned}
\vec{\nabla}_W \mathcal{F}(\Phi, \Theta) &= \sum_n \vec{\nabla}_W \big\langle \log \big( p_\Theta(\vec{x}^{(n)} \,|\, \vec{z}) \, p_\Theta(\vec{z}) \big) \big\rangle_{q_\Phi^{(n)}} \\
&= \sum_n \vec{\nabla}_W \big\langle \log \big( p_\Theta(\vec{x}^{(n)} \,|\, \vec{z}) \big) \big\rangle_{q_\Phi^{(n)}} = \sum_n \vec{\nabla}_W \big\langle \log \big( \mathcal{N}(\vec{x}^{(n)}; \vec{\mu}(\vec{z}, W), \sigma^2 \mathbb{I}) \big\rangle_{q_\Phi^{(n)}} \\
&= -\frac{1}{2\sigma^2} \sum_n \vec{\nabla}_W \sum_{\vec{z} \in \Phi^{(n)}} q_\Phi^{(n)}(\vec{z}) \, \|\vec{x}^{(n)} - \vec{\mu}(\vec{z}, W)\|^2 \\
&= -\frac{1}{2\sigma^2} \sum_n \sum_{\vec{z} \in \Phi^{(n)}} q_\Phi^{(n)}(\vec{z}) \, \vec{\nabla}_W \|\vec{x}^{(n)} - \vec{\mu}(\vec{z}, W)\|^2 \, ,
\end{aligned}
\tag{11}
$$

where the weighting factors $q_\Phi^{(n)}(\vec{z})$ are by using (3) and (1) given by:

$$
\begin{aligned}
q_\Phi^{(n)}(\vec{z}) &= \frac{p_\Theta(\vec{x}^{(n)} \,|\, \vec{z}) \, p_\Theta(\vec{z})}{\sum_{\vec{z}' \in \Phi^{(n)}} p_\Theta(\vec{x}^{(n)} \,|\, \vec{z}') \, p_\Theta(\vec{z}')} \\
&= \frac{\exp\big(-\frac{1}{2\sigma^2} \|\vec{x}^{(n)} - \vec{\mu}(\vec{z}, W)\|^2 - \tilde{\vec{\pi}}^T \vec{z}\big)}{\sum\limits_{\vec{z}' \in \Phi^{(n)}} \exp\big(-\frac{1}{2\sigma^2} \|\vec{x}^{(n)} - \vec{\mu}(\vec{z}', W)\|^2 - \tilde{\vec{\pi}}^T \vec{z}'\big)}
\end{aligned}
\tag{12}
$$

for all $\vec{z} \in \Phi^{(n)}$, where $\tilde{\pi}_h = \log\big(\frac{1 - \pi_h}{\pi_h}\big)$. Note that the $q_\Phi^{(n)}(\vec{z})$ are evaluated at the current values of the parameters $\Theta$, they are therefore treated as constant, e.g., for the gradient w.r.t. $W$.

It may be interesting to compare the gradient estimate (11) to the gradient estimate of conventional VAE training. For this consider a standard encoder given by an amortized variational distribution which we shall denote by $\tilde{q}_\Phi^{(n)}(\vec{z})$. The distribution $\tilde{q}_\Phi^{(n)}(\vec{z})$ could be a Gaussian whose mean and variance are set by passing data point $\vec{x}^{(n)}$ through encoder DNNs. For discrete VAEs, $\tilde{q}_\Phi^{(n)}(\vec{z})$ can be thought of as an analog discrete distribution. If we now take gradients of (6) w.r.t. $W$ and estimate

---

[1]For our choice of variational distributions, it is not trivial that the entropy term actually can be ignored because the encoding model $q_\Phi(\vec{z}; \vec{x})$ in (3) is defined in terms of the decoding model and its parameters. For truncated distributions, however, it can be shown that the entropy term can still be ignored (Lücke, 2019).

using samples from $\tilde{q}_\Phi^{(n)}(\vec{z})$, we obtain the familiar form:

$$
\begin{aligned}
\vec{\nabla}_W \mathcal{F}(\Phi, \Theta) &= \sum_n \vec{\nabla}_W \big\langle \log \big(p_\Theta(\vec{x}^{(n)} \mid \vec{z}) \, p_\Theta(\vec{z})\big) \big\rangle_{\tilde{q}_\Phi^{(n)}} \\
&= \sum_n \vec{\nabla}_W \big\langle \log \big(\mathcal{N}(\vec{x}^{(n)}; \vec{\mu}(\vec{z}, W), \sigma^2 \mathbb{I}) \big\rangle_{\tilde{q}_\Phi^{(n)}} \\
&\approx -\frac{1}{2\sigma^2} \sum_n \frac{1}{M} \sum_{m=1}^M \vec{\nabla}_W \|\vec{x}^{(n)} - \vec{\mu}(\vec{z}^{(m)}, W)\|^2 , \quad \text{where } \vec{z}^{(m)} \sim \tilde{q}_\Phi^{(n)}(\vec{z})
\end{aligned}
$$

We can slightly rewrite this expression to obtain:

$$
\vec{\nabla}_W \mathcal{F}(\Phi, \Theta) \approx -\frac{1}{2\sigma^2} \sum_n \sum_{\vec{z} \sim \tilde{q}_\Phi^{(n)}} \left(\frac{1}{M}\right) \vec{\nabla}_W \|\vec{x}^{(n)} - \vec{\mu}(\vec{z}, W)\|^2 , \tag{13}
$$

If we now compare with the gradient using the truncated approximation $q_\Phi^{(n)}(\vec{z})$,

$$
\vec{\nabla}_W \mathcal{F}(\Phi, \Theta) = -\frac{1}{2\sigma^2} \sum_n \sum_{\vec{z} \in \Phi^{(n)}} q_\Phi^{(n)}(\vec{z}) \, \vec{\nabla}_W \|\vec{x}^{(n)} - \vec{\mu}(\vec{z}, W)\|^2 , \tag{14}
$$

one can discuss analogous roles played by the subsets $\Phi^{(n)}$ (the variational parameters of $q_\Phi^{(n)}(\vec{z})$) and by a standard encoder $\tilde{q}_\Phi^{(n)}$. The states in a subset $\Phi^{(n)}$ are used to estimate the gradient similar to the samples from a standard encoder $\tilde{q}_\Phi^{(n)}(\vec{z})$. The size of $\Phi^{(n)}$ can consequently be thought of as analog to the number of samples used in a conventional estimation of the gradient. Standard VAE training estimates the gradient by weighting all samples equally (with $(1/M)$) and the gradient direction is approximated using sufficiently many samples drawn from the current $\tilde{q}_\Phi^{(n)}(\vec{z})$. In contrast, truncated gradient estimation uses the states in $\Phi^{(n)}$, and the gradient is computed using a weighted summation with weights $q_\Phi^{(n)}(\vec{z})$. These weights are computed by passing the states $\vec{z}$ through the *decoder* network. The gradient is then, notably, not a stochastic estimation but exact: gradient ascent is guaranteed (for small steps) to always monotonically increase the variational lower bound.

**Computational Complexity.** To add to the discussion of computational complexity of TVAE compared to standard VAE training, consider again Eqns. 13 and 14. If as many samples $M$ are used, per data point, as there are states in each $\Phi^{(n)}$, then both sums have the same number of summands. The evaluation of the gradients of the mean square error (MSE) is consequently precisely the same for both approaches. The additional weighting factors $q_\Phi^{(n)}(\vec{z})$ have to be computed for TVAE. However, the weighting factors just represent a small overhead because the evaluation of the decoder DNN for the states in $\Phi^{(n)}$ is a computation that can be reused from the updates of $\Phi^{(n)}$.

The main computational differences are in the updates of $\Phi^{(n)}$ compared to the update of encoder DNNs for conventional VAEs. Once the parameters $\Theta = (W, \sigma^2, \vec{\pi})$ are updated using (14), new states for $\Phi^{(n)}$ have to be sought based on criterion (9). In practice and for each $n$, we generate $M'$ new states according to the applied evolutionary procedure. To select the best states we have to pass all these $M'$ new states through the decoder DNN to evaluate (9). Furthermore, we have to pass all $M$ states already in $\Phi^{(n)}$ through the DNN to re-evaluate (9) because the parameters $\Theta$ have changed. In summary, we do require $\mathcal{O}(N \times (M + M'))$ passes through the decoder DNN. Selecting the $M$ best states from the $(M + M')$ states does not add complexity as this can be done in $\mathcal{O}(M + M')$ for each $n$ (Blum et al., 1973). The EA does add to the computational load but parent selection and mutation only add a constant offset for each of the considered states.

For comparison with standard VAEs, if we use $M$ samples of an encoder $\tilde{q}_\Phi^{(n)}(\vec{z})$, we require $\mathcal{O}(M \times N)$ passes through the decoder DNN to update the parameters $\Theta$ according to (13). For the encoder update, one requires $N \times \tilde{M}$ passes through encoder and decoder DNN to estimate the gradient w.r.t. the encoder weights (if we draw $\tilde{M}$ samples for each data point from a conventional encoder distribution $\tilde{q}_\Phi^{(n)}(\vec{z})$. The additional overhead to actually draw the samples is usually negligible.

Hence, the computational complexity of TVAE training is comparable if $M \approx M' \approx \tilde{M}$. However, conventional VAE training is amortized, i.e., the update of encoder weights uses information from all

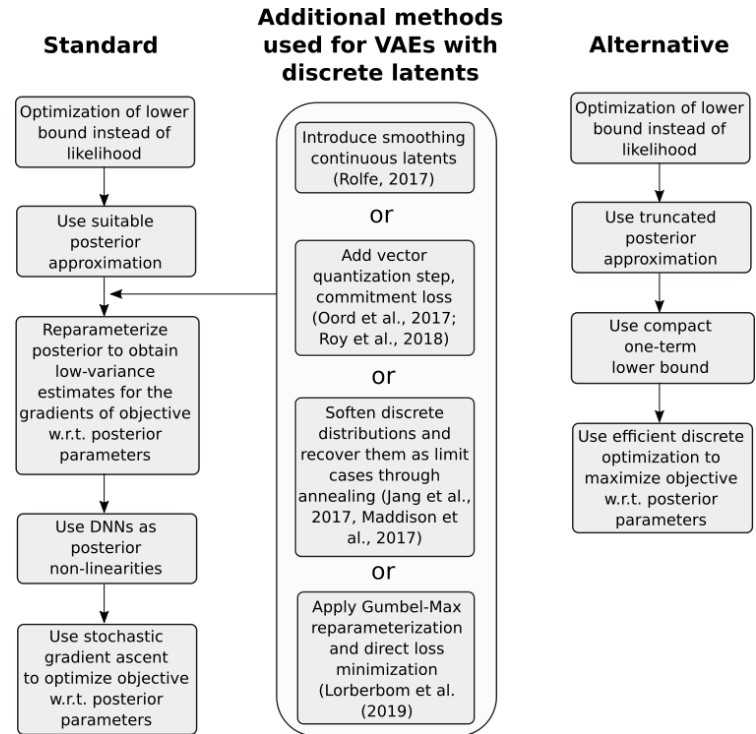

Figure 6: Standard series of methods applied to optimize the encoding model of VAEs. **Left:** methods applied for encoding models of standard VAEs. **Middle:** additional methods applied to maintain the standard procedure of encoding model optimization also for discrete latent variables. **Right:** alternative approach to optimize the VAE encoding model using direct discrete optimization.

data points $n$. In contrast, TVAE training is not amortized, i.e., the $\Phi^{(n)}$ are updated per data point. The advantage of amortization is that in practice, weights of a conventional encoder can converge faster or (alternatively) less samples $\tilde{M}$ are required. Considering the observed runtimes, more efficient conventional VAE training can presumably in large parts attributed to faster convergence using amortization. Furthermore, the used number of samples $M$ for conventional VAE training is usually smaller than best working sizes of $\Phi^{(n)}$ (we used, e.g., $|\Phi^{(n)}| = 64$ and $|\Phi^{(n)}| = 200$ for denoising, see Tab. 3); and the required storage of $\Phi^{(n)}$ results in overhead computations. On the other hand, amortization also has disadvantages (e.g. Kim et al., 2018; Cremer et al., 2018). The competitive performance for denoising may consequently be attributed at least in part to TVAE not being subject to an amortization gap.

## B    DETAILS ON THE NUMERICAL EXPERIMENTS

### B.1    VERIFICATION ON GROUND-TRUTH DATA

We first evaluated TVAE training on artificial datasets with known ground-truth parameters and log-likelihood, in order to verify the correct functioning of the algorithm and to investigate possible local optima effects. The dataset consisted of 500 4x4 images generated by linear superposition of vertical and horizontal bars, with a small amount of Gaussian noise. The DNN's input and middle layers had 8 units each. The $\Phi^{(n)}$ variational sets consisted of 64 hidden states each. Fig. 7 shows the evolution of the run that achieved the highest ELBO value out of ten. All parameters were correctly recovered, and the ELBO value was consistent with actual ground-truth log-likelihood.

Such a simple test, however, can also be solved by linear models. In order to demonstrate that TVAEs can solve non-linear problems, taking advantage of the neural network non-linearity embedded in the generative model, we introduced correlations between pairs of bars: the bars combinations shown

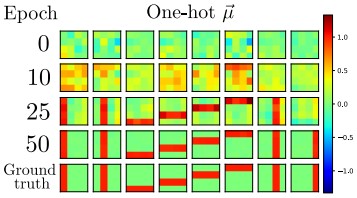

Figure 7: TVAE training on simple bars data: noiseless output of the TVAE's DNN for the 8 possible one-hot input vectors over several training epochs. Generating parameters are in the last row.

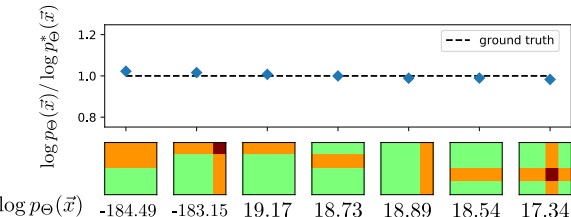

Figure 8: Correlated bars test. The plot shows the ratio between inferred and ground-truth log-likelihoods $\log p_\Theta(\vec{x})$ of datapoints with interesting bar combinations. The inferred values are reported below the datapoints themselves.

in the first two datapoints from the left in Fig. 8 were discouraged from appearing together. We employed the same evolutionary scheme and again we selected the run with highest peak ELBO value out of ten. The model correctly learns that certain combinations of bars are much more unlikely than others, and correctly estimates their likelihood.

Fig. 9 offers some more insight into the correlated bars test experiment described. The left section of the figure shows the generative parameters for the dataset used: $W_0$ is the 8x8 weight matrix of the top-to-middle layer: this makes it so that the activation of the first latent variable inhibits activation of the second, and activation of the last latent variable inhibits activation of the last. Concretely, this results in a dataset where these specific bars combinations are discouraged from appearing. The weights $W_1$, visualized as 8 4x4 matrices, generate the actual bars. $\sigma^2$ was set to 0.01 and the dataset contained an average of two superimposing bars per datapoint ($\pi_h = 2/8$ for each $h$).

The middle section of the figure shows the ELBO values (averages over all batches for each epoch) as training progresses. The cyclic learning rate schedule is responsible for the oscillatory behavior.

The right section shows some example datapoints together with samples from the trained TVAE model that reached the highest ELBO value out of the ten runs.

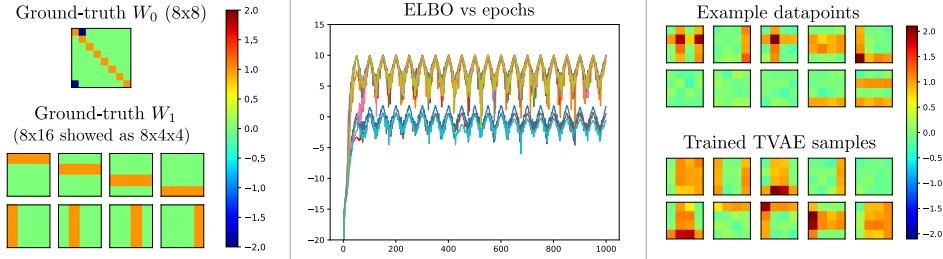

Figure 9: From left to right: generative parameters for the correlated bars test; ELBO values over epochs for 10 runs; example datapoints and samples from the generative model.

## B.2 DENOISING

Given a trained TVAE with parameters $\Theta$, we estimated the value of a pixel in a single patch as $x_d^{\text{est}} = \langle x_d \rangle_{p_\Theta(x_d | \vec{x})}$. When using $p_\Theta(x_d \mid \vec{x}) = \sum_{\{\vec{z}\}} p_\Theta(x_d \mid \vec{z}) p_\Theta(\vec{z} \mid \vec{x})$ we obtain:

$$x_d^{\text{est}} = \left\langle \langle x_d \rangle_{p_\Theta(x_d | \vec{z})} \right\rangle_{p_\Theta(\vec{z}|\vec{x})} = \langle \mu_d(\vec{z}) \rangle_{p_\Theta(\vec{z}|\vec{x})} . \tag{15}$$

The expectation value on the right-hand-side of Eqn. (15) is then approximated based on the encoding parameters $\Phi^{(n)}$ using truncated posteriors. Finally, we took a weighted average of the estimates of a pixel value in different patches (see, e.g., Burger et al., 2012) in order to generate the pixel values of the full denoised image.

In Tab. 3 we list the exact hyper-parameters used to obtain the PSNR values reported. In parentheses, the parameters for the run on data with noise level $\sigma = 50$ and unconstrained hyper-parameters are given, when they differ from the other experiments.

Table 3: Hyper-parameters for the denoising experiments on the house image.

| Neural network units | |
|---|---|
| Input ($H$) | 64 (512) |
| Middle | 64 (512) |
| Output ($D$) | 64 (144) |

| Cyclic Learning Rates | |
|---|---|
| Min l.r. | 0.0001 |
| Max l.r. | 0.01 (0.05) |
| Epochs/cycle | 20 |
| Batch size | 32 |

| Evolutionary parameters | |
|---|---|
| Parents | 10 (5) |
| Children | 9 (4) |
| Generations | 4 (1) |
| Size of $\Phi^{(n)}$ | 200 (64) |

Table 4: Denoising performance of n2v in PSNR (dB) for the 'house' image with AWG noise. For comparison, we additionally list the performance of TVAE (numbers copied from Tab. 2). PSNR values for n2v$^\star$ are obtained by training only on the noisy image (i.e., in the same setting as used for MTMKL, GSC, var-BSC and TVAE in Tab. 2. More training data improves performance for n2v. PSNR values for n2v$^\dagger$ show performance if additional training data in the form of noisy images with AWG noise $\sigma = 25$ is used. Further improvements (especially for high noise) are obtained if the n2v network is trained on training data with a noise level that matches the noise of the test set (see n2v$^\ddagger$). For instance, we used for n2v$^\ddagger$ training data with $\sigma = 50$ to denoise the 'house' with $\sigma = 50$. See text for further details.

| | $\sigma$=15 | $\sigma$=25 | $\sigma$=50 |
|---|---|---|---|
| n2v$^\star$ | 32.05 | 29.20 | 25.42 |
| n2v$^\dagger$ | 32.93 | 32.10 | 20.96 |
| n2v$^\ddagger$ | 33.91 | 32.10 | 28.94 |
| TVAE | **34.27 ± .02** | **32.65 ± .06** | **29.98 ± .05** |

To evaluate the performance on standard denoising benchmarks, we first compared TVAE to related probabilistic sparse coding approaches such as MTMKL, GSC and var-BSC (Tab. 1). MTMKL and GSC use the data model of spike-and-slab sparse coding and for training mean-field and truncated posterior approximations with pre-selection are used, respectively. Compared to MTMKL and GSC, var-BSC uses a less complex data model and a training scheme also based on evolutionary optimization (Guiraud et al., 2018). The denoising performance observed in the scenario with controlled conditions (Tab. 1) shows that for high noise level ($\sigma = 50$), var-BSC achieves higher PSNR values than MTMKL and GSC although the method uses a simpler data model. This observation demonstrates the effectiveness of the evolutionary training method used by var-BSC. However, PSNR values for TVAE are significantly higher due to the higher flexibility in modeling the data distribution provided by the used DNN.

In a second step, Tab. 2 compared the performance of TVAE with respect to different denoising approaches including deterministic sparse coding (KSVD), a mixture model approach (EPLL), a non-local image processing method (WNNM) and state-of-the-art denoising methods based on deep neural networks (BDGAN and DPDNN). These approaches can be distinguished, e.g., by the amount of employed training data and by the requirement for clean data.

TVAE as well as MTMKL, GSC and var-BSC do not require clean images for training. Furthermore, all these approaches can be trained if only the single noisy image is available ('zero-shot' learning; compare, e.g., Shocher et al., 2018; Imamura et al., 2019). Instead, EPLL, BDGAN and DPDNN use clean training data (typically tens or hundreds of thousands of data points are collected for training).

Approaches such as noise2noise (n2n Lehtinen et al., 2018) and noise2void (n2v Krull et al., 2019) occupy a middle ground: they can be trained on noisy data but they typically require much larger amounts of data than, e.g., TVAE or MTMKL. In the original n2v publication, for instance, 400 (noisy) $180 \times 180$ BSD (Martin et al., 2001) images were used to create a training dataset (this procedure also involved data augmentation; compare Krull et al. 2019). For our comparison with results of Tab. 2, we used the standard, publicly available code for n2v together with the default training set ($\sigma = 25$) employed in the original n2v publication. We then applied the trained n2v network to denoise the 'house' image with $\sigma = 25$. The resulting PSNR value was $32.10dB$ which is $0.76dB$ lower than the PSNR value for BM3D ($32.86dB$). The difference is consistent with an on average $0.88dB$ lower performance of n2v compared to BM3D on the BSD68 test set (see Krull et al., 2019). The same network can also be used to denoise an image with lower or higher noise level. The n2v network trained on $\sigma = 25$ does, for instance, result in PSNR values of $32.93dB$ for the 'house' image with $\sigma = 15$ and in $20.96dB$ for the 'house' image with $\sigma = 50$ (see $n2v^{\dagger}$ in Tab. 4). Especially for high noise levels performance can be much improved, however, if the n2v network is trained using images with the same noise level as the test image. In order to do so, we followed the procedure described in the n2v publication while adapting the noise level of $\sigma = 15$ in one case and $\sigma = 50$ for the other case. Trained on a dataset with matched noise, we then denoised the 'house' image with $\sigma = 15$ in the one, and $\sigma = 50$ in the other case (results listed as $n2v^{\ddagger}$ in Tab. 4). The PSNR values obtained for 'house' in this matched-noise-level scenario are much higher compared to the scenario with unmatched noise level (e.g., for $\sigma = 50$ the PSNR improvement is approximately 8 dB). The much lower performance for mismatched noise for n2v is in this respect consistent with observations for standard DNN denoising for which training with the ground-truth noise level has been pointed out as important for performance (Chaudhury & Roy, 2017; Zhang et al., 2018).

The n2v approach can avoid having to know the exact noise level, e.g., if it is trained on just the single noisy image. In a last experiment, we hence investigated this 'zero-shot' denoising feature of n2v and applied the algorithm to denoise the 'house' image while using the same noisy image for training that we seek to denoise (we took the publicly available code of n2v as an example and manually adjusted hyperparameters as follows: we set the "Percentage of pixel to manipulate per patch" to a value of 0.4, as "Number of training epochs" we used 400 and we set the "Number of parameter update steps per epoch" to 33). The obtained PSNR values are listed as $n2v^{*}$ in Tab. 4.

From Tab. 4 it can be observed, that for all considered training settings of n2v and all noise levels, PSNR values of TVAE are consistently higher than those of n2v even if n2v is trained on external data with matched-noise level. Additional parameter tuning may improve performance of $n2v^{*}$ to a certain extent but PSNRs are in general much lower than $n2v^{\ddagger}$. While we followed for $n2v^{\ddagger}$ the standard hyperparameter setting of the original paper/code publication of n2v (Krull et al., 2019), we cannot exclude further improvements with parameter fine tuning for the 'house' benchmark. However, we remark that the difference of $n2v^{\ddagger}$ and BM3D for the 'house' benchmark is on the very same range as the differences between n2v and BM3D as reported on the BSD data set in the original n2v publication. The stronger performing BM3D is according to denoising performance the preferable comparison and as such included in Tab. 2. In terms of efficiency, the n2v approach is in general (once trained) faster than BM3D as well as TVAE, however.

PSNR values of noise2noise (n2n) are usually very closely aligned with PSNR values achievable by feed-forward DNNs. More concretely, n2n uses, for instance, a RED30 network (Mao et al., 2016) which achieves 31.07 dB PSNR on the BSD300 data set if trained on clean data. If directly trained on noisy data, RED30 achieves 31.06 dB (Lehtinen et al., 2018). n2n is thus strongly performing in terms of PSNR. The caveat of n2n compared to n2v is, however, that the noisy data n2n uses is rather artificial. The pairs of images n2n is trained on consist of two different noise realization of the same underlying clean image. For real data, such a setting is only approximately occurring at most, which has motivated the n2v approach.

Like n2v, BDGAN and DPDNN are optimized for specific noise levels (specific standard deviations are used to generate the noisy training examples). EPLL is trained exclusively on clean image patches; for denoising, the algorithm requires the ground-truth noise level of the test image as input parameter. Ground-truth noise level information is also required by KSVD and WNNM.

Like all approaches in the top category of Tab. 2, TVAE does not require ground-truth noise level information, nor clean images, nor large amounts of training data. For the 'zero-shot' setting, TVAE

is consequently the best performing system on the 'house' benchmark. Such a high performance is notably achieved using a basic DNN and relatively small patch sizes of $D = 8 \times 8$ (for $\sigma = 15$ and $\sigma = 25$) or $D = 12 \times 12$ (for $\sigma = 50$). All feed-forward DNNs for denoising use much larger patches (e.g., n2v use $64 \times 64$). That a competitive denoising performance can be achieved for small patches, in general, argues in favor for VAE approaches to denoising. Indeed, TVAE even comes close to state-of-the-art approaches (BDGAN and DPDNN) that use very intricate DNN architectures and large amounts of clean training data. We believe that such results underline the potential of the here investigated approach although the novelty of the approach is the focus rather than extensive benchmarking.

On the other hand, an important limitation of TVAE is its computational demand. For our experiments on the 'house' image with noise level $\sigma = 50$ in Tab. 2 we used $N = 60025$ patches of $D = 12 \times 12$ pixels, which amounts to all possible non-overlapping square patches of that size that can be extracted from the image. For training and denoising we used a TVAE with $H = 512$ latent variables, sizes of $|\Phi^{(n)}| = 64$, and $512$ units in the DNN middle layer of the decoder. TVAE training required 49 seconds per training epoch when executing on a single NVIDIA Titan Xp GPU and 2.5 GB of GPU memory. We ran for $500$ epochs which required between seven and eight hours on the single GPU. We did not observe significant changes in variational bound values or in denoising performance after $500$ epochs in any of the experiments we conducted for Tabs 1 and 2. Runtime complexity increased linear with the number of data points $N$, with the dimensionality of the data $D$, with the number of the latents $H$, and with the size of the DNN used. Runtimes also increased approximately proportional w.r.t. the size of $\Phi^{(n)}$. Empirically we observed a sublinear scaling with $|\Phi^{(n)}|$ presumably because of significant overhead computations: for example, increasing from $|\Phi^{(n)}| = 64$ to $|\Phi^{(n)}| = 128$ (while keeping all other parameters as above) computational time increases from 49 seconds per training epoch to 75 seconds.

For noise levels $\sigma = 15$ and $\sigma = 25$ in Tab. 2 we used smaller patch sizes ($D = 8 \times 8$) and fewer stochastic latents ($H = 64$) but larger $\Phi^{(n)}$ (i.e., $|\Phi^{(n)}| = 200$). In general, if the patch size $D$ is increased, more structure has to be captured. This can be done either by increasing the size of the stochastic latents $H$ or by using larger DNNs. Both, in turn, requires more training data in order to estimate the increased number of parameters. In the current setup, the sizes of $D$ which are currently feasible are comparably small. The denoising performance based on small patches is, however, notably very high.

For comparison, n2v uses up to $D = 64 \times 64$ and also all other feed-forward DNN approaches use significantly larger patch sizes than TVAE (and the other approaches in category 1). Still, n2v can be trained efficiently on large patches requiring approximately 19 hours on a NVIDIA Tesla K80 GPU for training on approximately 3k noisy images of shape 180x180 and seconds for the denoising of one 256x256 image. The higher computational demand of TVAE is also the reason why averaging across databases with many images (such as BSD68) or applications to large single images quickly becomes infeasible. As a novel approach, TVAE is, however, far from being fully optimized algorithmically compared to large feed-forward approaches, and there is certainly further potential to improve training efficiency.

While denoising is, in general, well suited for deep generative models, performance for standard image denoising is by far not as common as such results for standard DNNs (which may also be related to efficiency aspects). An exception is a recent GAN approach (BDGAN; Zhu et al., 2019). VAEs are often evaluated using binarized MNIST with approximate log-likelihoods for comparison; that benchmark, however, is not consistent with the Gaussian noise model used here and does not allow a direct comparison with feed-forward DNNs which are the state-of-the-art.

