# OpenReview forum: "Direct Evolutionary Optimization of Variational Autoencoders with Binary Latents"
_ICLR.cc/2021/Conference — Reject_

### Official Review · AnonReviewer2 · 2020-10-27
**Interesting idea of using evolutionary search for training VAEs, analysis can be improved by comparing with existing approaches**

**Rating:** 6
**Confidence:** 3

**Review:**

**summary**
the paper proposes a novel approach to training variational autoencoder models, based on non-parametric form of truncated approximate posterior. Posterior is truncated to have support on a small subset of latent space allowing for exact marginalization. The support of approximate posterior in latent space for each data point $x$ is learned via evolutionary algorithm, minimizing ELBO. Method is applied to denoising tasks for images.

**pros**
the idea of using evolutionary algorithm for approximating truncated posterior is new to me, the fact that it works in practice is very interesting.

**cons**
* The idea is limited to small architectures due to the need to evaluate the decoder for every state in the truncated set as opposed to single evaluation in conventinal approaches. The authors for this reason choose to apply their approach to denoising applications, which I am not familiar with and can't evaluate the advantage of their method.
* The fact that approximate posterior is not amortized over the dataset is also a potential issue: for each data point $q(z|x)$ will need to be updated to match evolution of the decoder of the entire epoch. It seems plausible that applying more evolutionary steps can take care of this but it would be useful to see experiments with varying number of  evolutionary steps.
* authors didn't compare their approach to the recent paper "Efficient Marginalization of Discrete and Structured Latent Variables via Sparsity" https://arxiv.org/abs/2007.01919, it would be good to understand which one works better.


comments:
* The approach is reminiscent of EM approach where samples from the posterior are approximated via Metropolis-Hastings updates. This can be made more efficient by adding temperature annealing. I would be interested to see how the proposed approach compares to this.
* it would be clearer to make x-dependence in Eq (3) explicit rather then just through superscript (n)
* it would be interesting to see how large is gap between the proposed ELBO and true LL in the trained models, to get a sense of good the approximation is.

---

> ### Author Response · Authors · 2020-11-16
> **Authors' reply**
>
> Thank you for your review. The number of evolutionary steps taken per data point to optimize the variational parameters (i.e. the parameters of the encoder) can be controlled by the number of generations of candidate states evolved per variational E-step (compare Alg. 1; the number of generations of candidate states is a hyper-parameter of the algorithm). While the evolution of multiple generations turned out to be advantageous for denoising (Tab. 4), we did not observe a significantly different behaviour of the algorithm in the verification experiments (Sec. B.1) when varying the number generations evolved.
>
> Thank you for pointing out the nice paper by Correia et al., which indeed shares some similarities with our own. Correia et al. maintain a DNN-based posterior approximation and propose an efficient algorithm for setting most of the values of the discrete variational distribution to hard zeros. In contrast, our work is based on a mathematically rigorous variational lower bound that is a provably tight approximation of the true log-likelihood in sparse settings. Unlike Correia et al., our method does not require special care or further approximations in case of large latent spaces of exponential size, and the lower bound is guaranteed to never decrease as the sets of latent states are updated via Eqn. 8. An empirical comparison of the methods would certainly be interesting.
>
> As evolutionary algorithms do not get stuck on a single mode of the distribution as much as MCMC approaches do, our experience is that our method does not require simulated temperature annealing to perform adequately.
>
> Regarding the gap between the model's ELBO and the true LL: in the appendix we show that (at least for a small artificial dataset where the true LL is easily computable) we recover ELBO values consistent with the true LL value, suggesting that at least in principle the model can avoid amortization or approximation gaps.
>
> We will also update the paper with the notation improvements suggested.

---

### Official Review · AnonReviewer4 · 2020-10-28
**Novel and interesting idea with limited empirical evaluation**

**Rating:** 6
**Confidence:** 3

**Review:**

This paper proposes a new approach to train VAEs with binary latents, using an evolutionary algorithm to optimise a discrete set of variational parameters rather than the usual amortised variational model trained with gradient-based methods. The authors consider the setting of training a VAE with discrete, Bernoulli distributed latents and a continuous, Gaussian distributed output. They use a novel approach to form and train a posterior on the discrete latent space, parameterising the posterior somewhat implicitly via the sets defining a truncated posterior - in which the approximate posterior is proportional to the true posterior, but only within a subset of all points. Defining the subset of points amounts to parameterising the approximate posterior. Optimisation of these parameters amounts to a discrete search problem, which the authors tackle using an evolutionary algorithm.

Overall, I find the paper very well written and straightforward to follow. Indeed, the style of writing tends to pique the interest of the reader without being colloquial. I think the motivation of the paper is well established, given the number of papers that have investigated the training of modern generative models with discrete latents. Importantly, I think the proposed approach is novel and seemingly effective. The use of the truncated posterior along with a relatively simple evolutionary optimisation procedure is quite different from the previously seen methods that generally attempt to maintain a differential proxy to the ELBO.

The proposed method does have limitations, which the authors acknowledge. Namely that the method will not scale to large datasets, since the posterior is not amortised. I think it must also be true that the evolutionary optimisation is not particularly suited to high-dimensional spaces, since it is generally accepted that search suffers in such settings. However, these limitations I do not think are overwhelming. The authors find a niche problem setting in which their method is practical - that of “zero-shot” data denoising. In this setting they demonstrate that their method (with a small VAE) outperforms the state-of-the-art methods. The authors also provide a small set of straightforward and convincing toy experiments to show that their approach satisfies basic properties that we require in a learning system, such as appropriate scaling behaviour and recovery of artificial data generating parameters. Although I think the empirical demonstration may be enough, given the method is relatively novel, it is still limited. The main experimental result of the paper is denoising on a single image. Even performing the denoising on a small set of images would help the paper, since there will undoubtedly be variance in the performance across different images.

One other thing I would say about this paper is that identifying the approach as a VAE almost feels inappropriate. It is already true that the usual VAE is not an autoencoder at all, and so the name is a slight misnomer. But with the authors’ approach here, the posterior is not even parameterised as a function of the data (although there is of course an implicit dependence established through training). Hence identifying the method as related to an autoencoder seems to be misleading (although understandable given that the method is clearly related to the VAE).

---

> ### Author Response · Authors · 2020-11-16
> **Authors' reply**
>
> Thank you for your review. Please see our latest general comment for further motivation of the choice of benchmark, which (while acknowledging that further experimental results would strengthen the paper) we do not believe should be classified as "niche".
>
> Regarding the naming of the model: the declared intent is to juxtapose our approach to other methods proposed to deal with discrete latents in VAE-like generative models, compared to which we offer a more direct optimization procedure that naturally fits the discrete nature of the problem. Especially in a non-amortized setting, posteriors are indeed a function of the data, and the latent states corresponding to each datapoint are then fed into the decoding model in a manner not dissimilar to what happens in Variational Autoencoders.

---

### Official Review · AnonReviewer3 · 2020-10-29
**A different take on training VAEs**

**Rating:** 6
**Confidence:** 4

**Review:**

This paper proposes an evolutionary optimization framework for training vartional autoencoders (VAEs) with discrete latents. In contrast to the standard VAE paradigm, the proposed TVAE approach does not require an encoder for amortized inference given the input. The method instead relies on a pool of latent variable samples for each data point to activate the decoder network. The latent variable pools are maintained and iteratively updated to increase the average lower-bound of the marginal log-likelihood of the input data. Experimental results show that non-linear decoders optimized by the TVAE framework outperform their linear counterparts on a denoising task.  Further results demonstrate method's competitiveness on zero-shot denoising, where a TVAE decoder is only trained on the noisy input image to reconstruct a smoothed version of the input image.

The paper is well-written and easy to follow. The work proposes an interesting alternative for optimizing VAEs which does not require an encoder network for amortized inference. I however have a number of concerns that are as follows:

The method instead imposes the overhead of maintaining and evolving a collection of latent variables for every data point, which can be both sample inefficient and memory-heavy for sizable problems. Then when or why would one trade amortized inference for the proposed approach?

The paper falls short of comparing the proposed optimization procedure with other alternatives for training standard or discrete VAEs.

From reading the paper it is not clear how the size of the pool may need to be varied as the nature of the task or the number of latent changes.

The authors use a greedy approach to update the pool of latent samples. Does it not cause the optimization procedure to get stuck in local modes? Could one instead use MCMC type sampling approaches to enable mode jumps?

Why only denoising experiments? Can the authors use their approach for data synthesis tasks where the latents can be shown to control different attributes of the data generative process?

---

> ### Author Response · Authors · 2020-11-16
> **Authors' reply**
>
> Thank you for your review, please find our replies below.
>
> "when or why would one trade amortized inference for the proposed approach?"
>
> While true that non-amortized inference is more computationally demanding than amortized alternatives, it has been pointed out that the amortization gap [1] might result in sub-optimal inference; therefore non-amortized approaches might be preferable when the problem dimensionality allows it and a better approximation of the true log-likelihood might result in better performance (as it seems to be the case for zero-shot denoising).
>
> "It is not clear how the size of the pool may need to be varied as the nature of the task or the number of latent changes."
>
> Assuming a sparse data representation is possible (as it is often the case), the size of the pool does not need to vary much as a function of the dataset: it only needs to contain high-posterior latent states to be able to capture most of the posterior mass (and therefore deliver a good approximation of a potentially complex and multi-modal posterior structure). Empirically, for a fixed dataset, it can be verified that learnt free energy values as a function of the pool size typically plateau at sizes much smaller than the exponentially large maximum size; this same simple experiment can also be employed to pick a reasonable pool size for a given application.
>
> "The authors use a greedy approach to update the pool of latent samples. Does it not cause the optimization procedure to get stuck in local modes? Could one instead use MCMC type sampling approaches to enable mode jumps?"
>
> Note that the search procedure as discussed already enables mode jumps in principle: fitness-proportional parent selection and random uniform bitflips do not greedily approach the closest local optimum but could also find new modes. However, indeed MCMC-like sampling of proposal states would also be possible and is explored e.g. in [2].
> Empirically, evolutionary algorithms proved to perform just as well or better for our purposes.
>
> "Why only denoising experiments? Can the authors use their approach for data synthesis tasks where the latents can be shown to control different attributes of the data generative process?"
>
> See our general reply for further motivation of our benchmark setting. We are also working on inpainting applications when training on images with missing pixels, which is a task that our approach can deal with naturally, while e.g. traditional autoencoders must deal with missing pixels when passing datapoints through the encoding model. We could link to such results as supplementary material when available.
>
> [1] Cremer et al., 2018, "Inference suboptimality in variational autoencoders"
> [2] Lücke et al., 2018, "Truncated Variational Sampling for 'Black Box'Optimization of Generative Models"

---

### Official Review · AnonReviewer1 · 2020-11-02
**Ok theory; Clarity can be improved; application impact need to considered**

**Rating:** 5
**Confidence:** 3

**Review:**

This paper proposes to use evolutionary algorithm to learn truncated deep latent variable model. The method get good performance in denoising  task.

Pros:
Quality: The method seems correct
On denoising task, the performance of the proposed model is good.
Significance: Inference of Discrete VAE is an important question to address

Cons & questions:
Clarity: The paper is very hard to read due to many reasons:
a) It does not use commonly used notations, such as the paper use F to present ELBO. The paper seems to use theta to present the decoded mean and variance where people commonly use theta for the decoder weights etc. Under such unique notations which is not wrong, but the author did not explain them clearly either. Around equation (2), it just says that these are parameter to optimize (which made me think that these are the weights) and later on I found that weights are denoted by W.  and phi is the Bernoulli parameter for z (which was denoted as pi before) and theta is the decoded mean and variances. So it just makes reading very confusing.
b) It is not self contained. For example equation (7) is just pointed to another paper which I checked another paper for mins and did not find equation (7)  and did not try more as the cited paper is very very long. So it would be good to clarify critical equations.
I also wonder about Eq.(7)'s correctness as the right hand side looks like the model evidence instead of the lower bound of it.
Other places, such as Eq.(3) it is formulated in this way in the Truncated VAE paper but not cited.
Generically, the reading of the paper is not easy and there are many simple things to do just to make it more clear.

Significance:
a) The author seems try to claim that people in the field never used non-amortized way to train a deep latent Gaussian model. This is very wrong, also the author did not discuss related work in this direction at all. e.g. using MCMC
Learning Deep Latent Gaussian Models with Markov Chain Monte Carlo, http://proceedings.mlr.press/v70/hoffman17a/hoffman17a.pdf

b)  VAE's contribution are two-fold, one is latent variable model and the second part if the amortized inference. The paper does not seems differ such existing separation. The paper in the end contributed a non-amortized inference method, but the discussion of the paper is very entangled.

c) As the author pointed out, the proposed method is not scalable  as it is per data point and thus it will have limited application impact.

d) Experiments did not compare any highly relevant inference baselines. It does not compare any other inference methods in deep latent variable model.

e) there is only one set of experiments in total and on an traditional denositing task which is very limited. More experimental settings are needed to show the usefulness of the method.

---

> ### Author Response · Authors · 2020-11-16
> **Authors' reply**
>
> Thank you for your review. Regarding clarity: the use of "F" to denote the variational lower bound and \Theta to denote the generative model's parameters stems from Neal & Hinton ([1]) a very well known paper on the variational lower bound.
> There is a large body of literature that has since used the same notation, e.g. [2] for a recent example.
> Eqn. 7 is taken, as stated, from [3]. In that paper it is Eqn. 6. [3] may be long but it contains a three-page "Summary of the Algorithm" (Sec. 2) where Eqn. 7 appears. Eqn. 7 is sound, has been used in other papers [4,5] and was numerically verified here and in those other papers.
> We will, of course, improve on the notation where possible. At the same time, we feel encouraged by the other reviewers regarding the explanation of the method.
>
> Also note that:
> - we do not claim to be the first to train DLGMs in a non-amortized way; rather, we believe our original contribution is addressing discrete latents in a more direct fashion than previously investigated approaches, also side-stepping very commonly used techniques such as reparameterization and sampling approximations
> - although the method is more computationally demanding than alternatives, saying it is "not scalable" might be too strong of a statement given the discussed linear scaling with respect to the dataset size and the fact that memory and computational load can be distributed across machines
> - regarding the relevance of our experiments, please see our general reply as well as our reply to reviewer 3.
>
> [1] Neil & Hinton, 1998, http://www.cs.toronto.edu/~radford/ftp/emk.pdf
> [2] Vertes & Sahani, 2018, https://papers.nips.cc/paper/2018/file/955cb567b6e38f4c6b3f28cc857fc38c-Paper.pdf
> [3] Lücke, 2019, https://arxiv.org/abs/1610.03113
> [4] Forster et al., Neural Comp., 2018
> [5] Lücke et al., Pattern Recognition Letters, 2019

---

### Author Response · Authors · 2020-11-16
**Regarding our choice of experiments**

We thank the reviewers for their thoughtful feedback.
We would like to address the comments regarding our choice of denoising as benchmark for the novel training algorithm proposed.

As reviewer 4 stated, we investigated application domains where our approach can be competitive despite its discussed limitations (as is customary with new approaches, we would say). However, we would not so much say that zero-shot denoising is a niche: problems where just noisy data (or little data) is available are (A) a standard denoising setting, and (B) frequently and naturally occurring (low-light vision, electron-microscopy images etc.). It just so happens that for standard large NNs large clean datasets are required, which made that setting popular. Our use of relatively small networks is also the reason why we could not directly compare to other approaches: e.g., [1] used large networks of 18-20 layers and thousands of latent variables to obtain the competitive MNIST log-likelihood values reported. For a direct comparison with our approach, smaller DNNs would be required, meaning that we would have to repeat all experiments with smaller networks. That would have provoked criticism of us performing such experiments at scales biased towards our approach. The alternative choice that we made was to use published results which are readily available for the denoising task considered, which guarantees a fair comparison on equal grounds. The task also allowed for comparison with a broader range of methods including very recently suggested DNN-based denoising methods (e.g., [2]). We will further clarify in the paper the reasons for us deciding for the benchmark of zero-shot denoising.

[1] Rolfe, 2016, "Discrete Variational Autoencoders"
[2] Krull et al., 2019, "Noise2void-learning denoising from single noisy images"

---

### Author Response · Authors · 2020-11-24
**Response to all reviewers**

Following the point about additional experiments raised by the reviewers, we have included an inpainting benchmark and comparison with other recent approaches. Using TVAE with a standard feed-forward DNN (three layers, no convolutions, 500k parameters) we obtain a PSNR of 38.56 dB on the standard house benchmark with 50% missing pixels. In comparison, prominent and recent approaches achieve 38.02 dB (BPFA; Zhou et al., 2012), 34.58 dB (Papyan et al., ICCV, 2017) and 39.16 dB (DIP; Ulyanov et al., CVPR, 2018). The TVAE network is like BPFA and Papyan et al. (2017) itself permutation invariant while DIP uses convolutional layers and thus prior information about the 2D nature of images. We also remark that DIP relies on a two million parameter dedicated hour-glass DNN with LeakyReLu units and many optimized hyperparameters. We have uploaded the inpainting experiments to the supplementary material and will report the experiments in a revised version of this manuscript. We hope that these changes address the point raised and agree that including the experiments further improves the submission.

---

### Decision · Program_Chairs · 2021-01-07
**Final Decision**

**Decision:**

Reject

**Comment:**

The paper presents an evolutionary optimization framework for training discrete VAEs, which is different to the standard way of training VAEs. One of the main criticism of the paper was the choice of experiments, but the authors addressed this point by adding an inpainting benchmark.

Unfortunately, the reviewers' scores are borderline, and one of the reviewers pointed out the lack of scalability (more precisely, linear scalability with the number of observations) and cannot recommend acceptance based on the limited application impact. Given the large number of ICLR submissions, this paper unfortunately does not meet the acceptance bar. That said, I encourage the authors to address this point and resubmit the paper to another (or the same) venue.